# Program Report: Improving Patient Experience at an Outpatient Clinic Using Continuous Improvement Tools

**DOI:** 10.3390/healthcare11162301

**Published:** 2023-08-15

**Authors:** Muhammad Usman Hassan Siddiqui, Abdullah Ahmed Khafagy, Faisal Majeed

**Affiliations:** 1My Clinic International Medical Company, Prince Sultan Road, Jeddah 21411, Saudi Arabia; 2Department of Community Medicine and Pilgrims Healthcare, College of Medicine, Umm Al-Qura University, Mecca 24381, Saudi Arabia; aakhafagy@uqu.edu.sa; 3Occupational Health & Safety Compliance at Reckitt USA, Parsippany, NJ 07054, USA; faisal.majeed@reckitt.com

**Keywords:** hospital management, tertiary care, health care, administration, patient experience

## Abstract

Patient satisfaction with prompt and high-quality healthcare services plays a pivotal role in healthcare settings. The delivery of high-quality services within the healthcare sector is closely associated with continuous quality improvement (CQI), which is an incremental and progressive process that prioritizes the safety of all participants, favorable outcomes, systematic processes, and a regulated and improved working environment, particularly in later stages. Surprisingly, these aspects are less frequently explored in Middle Eastern countries. Thus, this research paper aims to assess the impact of quality services on patient satisfaction in tertiary care clinics located in the Middle East. To improve the quality of services in our clinic, we employed patient feedback as a valuable resource. We proactively reached out to all patients who had visited our hospital via mobile phone messages and requested their feedback on the services they received. Approximately 5% of all visitors responded and completed a comprehensive questionnaire. The majority of respondents expressed satisfaction with the services provided across various departments. However, they also offered valuable suggestions that helped us identify further areas for improvement and enhance the overall patient experience within our clinic. Drawing upon the feedback received, we meticulously considered the identified issues, redesigned our policies, and implemented strategic changes. Following the implementation of these new approaches, we once again sought patients’ feedback on the quality of our services. Patient feedback highlighted the significant impact of optimized service delivery methods, resulting in a substantial increase in patient satisfaction. Overall, this study sheds light on the vital factors that can enhance patients’ experience in outpatient clinics, emphasizing the importance of integrating patient feedback into continuous quality improvement initiatives. By utilizing this approach, healthcare providers, administrators, and researchers can effectively improve service quality and patient satisfaction. Consequently, this research paper serves as a valuable reference for public health stakeholders, administrators, and researchers in their pursuit of delivering exceptional healthcare experiences.

## 1. Introduction

In today’s modernized healthcare system, characterized by globalization, heightened patient expectations, increasing demands, and a fast-paced competitive environment, the implementation of continuous improvement strategies in healthcare setups has become imperative. A crucial aspect of optimizing the system with high efficiency lies in identifying gaps through quality indicators [1]. Traditionally, the selection of quality indicators for healthcare services has been a global concern. However, in the current value-based era, patient experiences have emerged as a pivotal indicator of quality healthcare [2]. The extensive literature on continuous quality improvement (CQI) emphasizes that well-organized, planned, and systematic approaches form the backbone of efficient organizations [3]. Additionally, the ability to swiftly manage patient flow and deliver high-quality work in clinics plays a vital role in minimizing waste and associated costs [4], ultimately alleviating the economic burden on customers.

The key to this process lies in the implementation of efficient and effective strategies, supported by committed stakeholders. Equipping staff with technical skills and empowering them to promote health behavior change is equally crucial in achieving organizational goals and fostering an efficient system [5]. Furthermore, system optimization is achieved through process optimization, value-added process mapping, problem identification and isolation, root cause analysis, and a meticulous approach to resolving issues [1]. It is important to note that various processes have been developed over time, but their efficacy has often been overlooked despite their efficiency in service and product provision. Identifying and optimizing wasteful and costly processes are essential to maintaining high-quality and goal-oriented standards in the industry [1]. In recent decades, researchers have dedicated efforts to improving processes by examining practices and drawing insights from various industrial settings [6].

CQI approaches in healthcare settings align with best clinical guidelines, encompassing aspects such as the availability of comprehensive services under one roof, optimal regularity of patient/client attendance, patient experiences leading to feelings of satisfaction or dissatisfaction, patient bedding, hospital waste management, and regular audits. However, it is essential to recognize that hidden factors may significantly impact the quality of services provided [1,7]. Despite this, there is a scarcity of studies utilizing patient feedback to enhance the quality of services. Thus, this study aims to bridge that gap by seeking patient feedback and analyzing their perspectives, which can serve as valuable insights for overcoming obstacles encountered at various stages of healthcare delivery.

## 2. Methods

### 2.1. Data Collection

This research was conducted within a multi-specialty outpatient clinic located in a multi-story building. Patients have the option to book their appointments either through a phone call or by visiting the clinic in person, where the staff collect their relevant information and prepare their file. To confirm the appointment, a Short Message Service (SMS) is sent to the patient’s mobile number. Upon arrival at the clinic and verification of their identity, the nursing team can access the patient’s status for triage. 

Following triage, the patient is called in by the physician for consultation. The entire clinic visit process is depicted in Figure 1, illustrating the overall design. Upon completion of the comprehensive checkup, patients are sent a message containing a link through which they can provide their feedback. However, the response rate from patients is relatively low, at approximately 5%. The data derived from patient feedback, based on the completed questionnaires, are presented in the subsequent figures. Continuous improvement tools were employed to further investigate the causes and effects of various factors and to identify the root causes and appropriate actions for improving the patient experience.

### 2.2. Continuous Improvement Process

The responses obtained from the patients’ feedback were meticulously recorded, enabling us to identify areas where issues were encountered. Subsequently, a comprehensive plan was devised to address these problem areas and enhance overall patient satisfaction. After a period of three months, the satisfaction of patients was re-evaluated to gauge the effectiveness of the implemented optimizations. The results obtained from the patients’ feedback are presented as percentages, providing a clear understanding of the level of satisfaction achieved. In order to gain deeper insights into the underlying causes of these issues, a fishbone diagram was developed, drawing upon indications from previously reported studies [8]. This diagram served as a valuable tool for comprehending the multifaceted factors contributing to the identified problems and helped guide our improvement efforts.

## 3. Results

### 3.1. Customer Satisfaction

As indicated in the existing literature, patient satisfaction stands out as the most crucial factor in driving continuous quality improvement efforts. While reasonable levels of patient satisfaction were reported, patients also provided valuable suggestions for further enhancement. The measurement of customer satisfaction (CSAT) was carried out through patient-filled questionnaires, which encompassed various touchpoints such as appointments, facilities, reception, nursing staff, clinics, pharmacy, laboratory, radiology, and insurance services.

The data obtained from the questionnaires reveal that patients reported the highest satisfaction with insurance services, with a CSAT score of 88%. This was followed by a CSAT score of 86% for facility-related aspects and the services provided by the clinic’s nursing staff. The CSAT score for clinic services and the radiology department was reported at 85%. Reception and appointment services received a CSAT score of 83% and 82%, respectively. However, the pharmacy department recorded the lowest satisfaction level, with a CSAT score of only 78% (see Figure 2).

Analysis of the feedback provided by all respondents who completed the questionnaire highlighted concerns related to the cooperation of physicians or surgeons at the clinic as the primary issue. Furthermore, respondents raised concerns about system-related issues, including the lack of display of expected waiting times and system bugs when additional requests were made. At the pharmacy, patients expressed concerns regarding the unavailability of prescribed medications and service codes. Additionally, patients noted the inefficient use of the queuing system. In the laboratory, although patients expressed overall satisfaction, they suggested the need for nurse education and involvement in proper training to improve their skills.

No specific suggestions were received for the radiology department and insurance stations. Nonetheless, we continued to work on improvements by taking into account the feedback and suggestions from other departments, ensuring the functionality of the continuous quality improvement program for the provision of superior healthcare services. A detailed framework, satisfaction reports, and patients’ feedback are provided in Figure 3 below, providing a comprehensive overview of the findings and the subsequent actions taken.

### 3.2. Root Causes of Patients’ Dissatisfaction

To identify the root causes of the problems encountered, we utilized a fishbone diagram as a visual tool, drawing insights from previous studies [8,9]. The diagram encompassed four key areas: system, process, people, and organization, with each shedding light on distinct root causes.

Within the system domain, we identified system bugs and restrictions as primary issues. System bugs included the absence of worklist updates with the receptionist and the failure to auto-adjudicate follow-up records. System restrictions encompassed challenges with the Que system software, the lack of waiting time display, and the absence of SMS updates regarding request status.

Regarding the process, two types of obstacles were identified: communication gaps and the prior approval process. Communication gaps arose due to a lack of detailed explanations provided by physicians or surgeons, constituting a common concern. Additionally, insufficient communication between the pharmacy and the insurance department was reported. The prior approval process was found to be non-automated, cumbersome, and resulted in unnecessary documentation (see Figure 4).

At the people level, cultural concerns, a lack of knowledge, and system misuse were observed. These system misuses included the failure to update approvals promptly in the system and improper or inadequate sealing of medical prescriptions for the pharmacy’s medication issuance.

Organization-level concerns fell into two categories. Firstly, resource availability and inefficient utilization of staff, coupled with inadequate patient management, were identified. These concerns encompassed issues such as the unavailability of prescribed medicines in the pharmacy, the lack of consistent service codes, the absence of an insurance office, and short dispensing durations (less than one month). Secondly, human resource management issues, including a lack of knowledge regarding outpatient clinic policies, were also noted (see Figure 4).

### 3.3. Continuous Improvement Program Application

Ensuring a superior patient experience is a paramount expectation from healthcare providers, as highlighted in previous studies [10,11]. To meet this expectation, we developed policies and implemented actions based on patients’ feedback, in alignment with the concerns identified through internal assessments. The primary focus was on improving patient satisfaction through targeted interventions.

The modified policies encompassed several key areas, including optimizing the utilization of human resources through comprehensive training, updating the system, establishing a monitoring setup, and conducting more frequent training workshops. A condensed overview of the identified points can be found in Table 1, providing further details.

During the initial stages of system optimization, we encountered several obstacles. These challenges included the need for efficient utilization of the system, the provision of round-the-clock support services, conducting training workshops to familiarize staff with the optimized system, and managing human resources effectively. However, we successfully overcame these obstacles by implementing a strategy that involved alternate turns of services for initially trained staff, proving to be instrumental in mitigating the challenges faced.

### 3.4. System Display and Preliminary Evaluation 

After the successful implementation of the project, a robust process was established to set key performance indicators (KPIs) and implement monitoring mechanisms. This process ensured that performance would be consistently measured, communicated, and any deviations would be identified for improvement. The system’s main cell display, depicted below, provides a live dashboard showcasing the number of patients at different stages of the process and the flow of work within this setup. Additionally, real-time monitoring enabled the calculation of percentages of patients at various points and time intervals, as illustrated in Figure 5.

Following the implementation of new policies and actions for a period of three months, we conducted a reassessment of patient satisfaction based on patients’ feedback. The analysis revealed notable improvements in patient satisfaction across various departments. Specifically, satisfaction in the pharmacy increased from 78% to 89%, reception satisfaction rose from 83% to 88%, and nursing satisfaction saw an increase from 92% to 96%. Likewise, higher levels of patient satisfaction were observed in the laboratory and radiology departments, with scores reaching 88% and 90%, respectively, compared to previous scores of 83% and 85%. Patient satisfaction in the clinic remained relatively stable, while satisfaction with insurance fell sharply from 98% to 78% after one month, followed by a gradual increase over time (see Figure 6). These findings indicate an overall improvement in patient satisfaction.

## 4. Discussion

### 4.1. Progress Made

Collectively, this study demonstrates the effectiveness of patient feedback-driven policies in enhancing the overall patient experience at outpatient clinics and significantly impacting patient satisfaction. Furthermore, the implementation of system automation, countercheck mechanisms, and real-time updated technologies is indispensable in seamlessly connecting clinics, diagnostic laboratories, and pharmacies within the outpatient setting. The short-term evaluation of these interventions has revealed a tangible increase in patient satisfaction.

### 4.2. Limitations

Considering the limitation of low response rates and the relatively short duration of evaluation following the implementation of revised policies and updated services, it is important to acknowledge that the long-term data results may not align precisely with the findings from the three-month period. To mitigate the potential ambiguity arising from this limitation, we have devised a plan to evaluate the patient feedback results after a specified timeframe. In this subsequent evaluation, we intend to explore additional factors, including insurance type, and incorporate patient satisfaction to provide a more comprehensive understanding of the survey outcomes.

## 5. Conclusions

Executive bodies across different business sectors continue to undertake initiatives to meet ever-changing demands in this volatile and uncertain business world and rapid technological developments. Similarly, our adoption of a continuous improvement process within the patient setting yielded positive outcomes. Encouraging patients to provide feedback proves instrumental in enhancing the quality of services delivered at healthcare facilities, ultimately contributing to improved registered outcomes [12].

## Figures and Tables

**Figure 1 healthcare-11-02301-f001:**
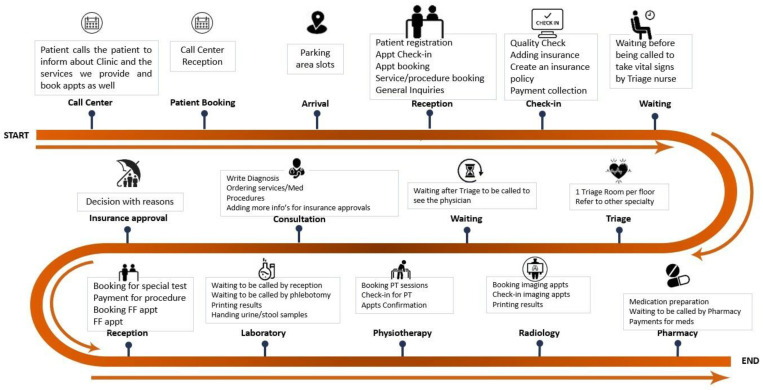
Flow chart of the process for patients’ services at outpatient clinic.

**Figure 2 healthcare-11-02301-f002:**
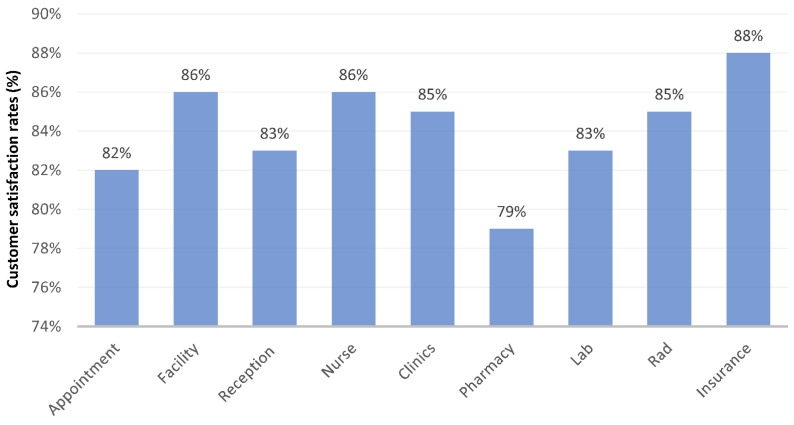
Customer satisfaction rates in different departments.

**Figure 3 healthcare-11-02301-f003:**
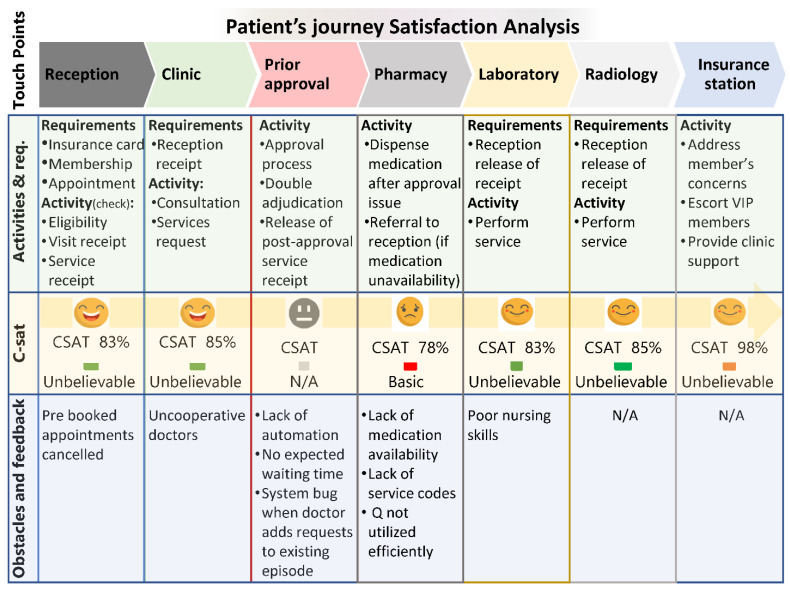
Combined overview view of touchpoints, activities, customer satisfaction, and reported obstacles.

**Figure 4 healthcare-11-02301-f004:**
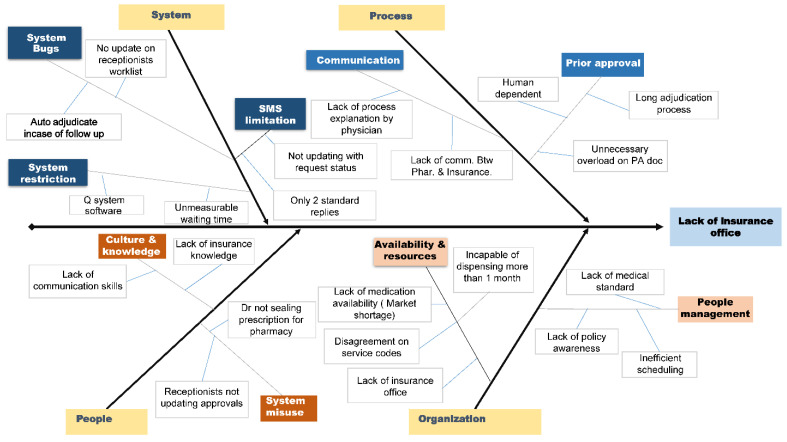
Problems identified in fishbone diagram.

**Figure 5 healthcare-11-02301-f005:**
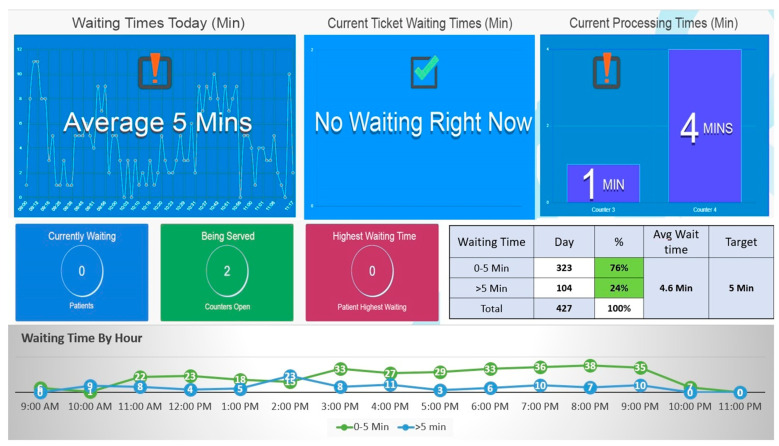
Real time display and frequencies of patients at different stages of visit along with time.

**Figure 6 healthcare-11-02301-f006:**
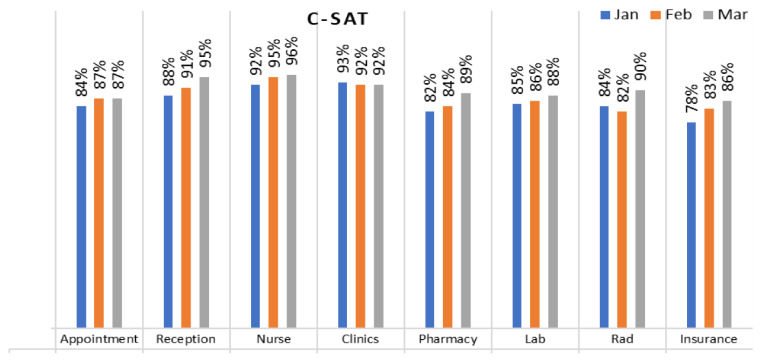
Post-updated action implementation patient satisfaction.

**Table 1 healthcare-11-02301-t001:** Implemented policies after first phase patients’ feedback and solutions with rigor.

Category	Countermeasure	Task	Priority	
People	Optimize the utilization of human resources	Establishing a task force dedicated to providing comprehensive training on the table of benefits to all relevant staff members	M	Done
		Implement technical skills training to ensure optimal utilization of the system, with a particular emphasis on providing doctors with access and proper training to effectively prescribe medications, thereby reducing the occurrence of technical errors and minimizing delays in the pharmacy.	H	Done
		Ensure compliance with prescribing requirements to ensure the completion of prescriptions (whether manual or electronic) before reaching the pharmacy.	H	Done
		Ensure effective communication between the pharmacy and the medical team to facilitate the exchange of any required information or changes at both levels. Additionally, service coordinators should provide support in cases where a response is not received	M	Done
		Provide comprehensive soft skill training to the entire pharmacy team, focusing specifically on customer service excellence	H	Done
		Implement a technical training program covering all retail products	H	Done
System	Eliminate system errors and enhance software utilization	Develop Queuing system software	H	Done
		Develop full patient journey monitoring	H	Done
		Revise the “MY Operation” system to eliminate incorrect auto-adjudication during follow-up episodes.	H	Done
		Verify the system’s capability to support the setup of the table of benefits.	M	Done
		Implement a system flag to identify uncovered medications at the point of delivery.	H	Done
		Integrate the system at the pre-authorization level to reduce processing time, mitigate human errors, and optimize the staffing needs for the insurance office at My Clinic.	M	Done
		Establish comprehensive SMS communication with members to provide updates on the status of their requests.	L	Done
		Pharmacy/Integration of Clinical Support software: to Integrate the Lexicomp/UpToDate with pharmacy & Clinical AppLonger dispensing time to ensure Patient safety (Drug Interactions/drug dosing safety…etc.)—Impact 3 Min web browsing	H	Done
		Implement automated barcode scanning to eliminate errors associated with batch barcode reading, ensuring accurate selection from a drop-down list in both the clinical application and point of sale for all items. This enhancement promotes patient safety by enabling the selection of the same dispensed batch and increases the dispensing time, resulting in a positive impact of one minute.	V.H	Done
		Integrate credit/debit card readers into the point of sale (POS) system at the pharmacy to facilitate automated payments for mixed self and insurance payments. This integration will streamline the payment process, reducing the time required for transactions by two minutes.	V.H	Done
Organization	Leveling and optimizing resources	Address service code issues in the pharmacy	V.H	Done
		Establish key performance indicators for clinical practice.	H	Done
		Allocate dedicated administrative staff to handle pre-authorized transactions exclusively.	M	Done
		Evaluate and revise the capacity plan for pharmacist scheduling.	M	Done
		Design and share the end-to-end process for the insurance office.	V.H	Done
Process	Automat back-end insurance office	Ensure timely procurement of all unavailable medications to maintain a consistent supply.	H	done
		Streamline inventory management practices for medications.	H	Done
		Revise the pharmacy planogram to enhance the display of newly added retail stock-keeping units, thereby boosting retail sales.	H	Done

Abbreviation: V.H: very high or top priority; H: high priority; M: medium priority; L: low priority.

## Data Availability

Data are contained within the article in different forms and can be requested as well in other formats through the corresponding author.

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
