# Peer review of "Program Report: Improving Patient Experience at an Outpatient Clinic Using Continuous Improvement Tools"

_healthcare, 2023, doi:10.3390/healthcare11162301_

Round 1
Reviewer 1 Report
1. The structure of this manuscript is very disorganized. I am not sure the contents of introduction, method, results and discussion are all in the right sections.
2. The authors need to explain the differences between patient experience and patient satisfaction in the introduction section. Did you really study the patient experience as described in the title? There are other quality issues being assessed in the study.
3. I am not surprised the response rate is only 5% in this type of survey. However, the system was revamped based on a 5% response rate. The authors need to discuss the potential bias or limitations arising from a system developed based on the feedback from a small proportion of patients.
4. The study (survey) results are not presented in the manuscript
5. The authors need to show how the mobile questionnaire is designed, pretest and the process of distributed the survey. The authors did not reveal the patient profile of those who responded to the original survey, the follow up survey after revamping the system as well as the response rates of the follow up surveys. A case in point, I am not sure the data sources used to support the claims by the authors that patient satisfaction increased from 78% to 89% within just 3 months . (line310) Also, the authors should conduct the basic statistical analyses to substantiate the positive result statements.
6. Quite a few topics presented in the manuscript are opinionated contents or the description of an observed process. For example, I am not clear that the data or survey results or rationale supporting the Table 1. I am not sure how you determined the priority column in the Table.
7. Describe the process of how the authors conducted the qualitative research to summarize the results. The fishbone diagram development process should be described in detail in the method section. The authors should articulate how the key waste areas and the problems were identified based on the survey results.
Author Response
Reviewer 1.
- The structure of this manuscript is very disorganized. I am not sure the contents of introduction, method, results and discussion are all in the right sections.
Authors response:
Dear Professor, we are highly grateful for your time, consideration and giving us an opportunity to revise the paper, re-organize it contents, improve the quality, minimize the existing mistakes and optimize the language to be able to publication overall. As far as, the contents of manuscript such as introduction, method, results and discussion are arrangement is taken from previously published papers and we believe is according to the journal requirement. Further, we would be highly grateful if you can advise us a better format, we will follow that as per your recommendation. After, your affirmative comments, we have carefully revised the manuscript and addressed such issues one by one and responses are provided with green highlighted text followed by each comment one by one. Moreover, while revising the manuscript we have activated track changes mode and highlighted newly inserted text in green background as well.
- The authors need to explain the differences between patient experience and patient satisfaction in the introduction section. Did you really study the patient experience as described in the title? There are other quality issues being assessed in the study.
Authors response:
Dear reviewer, we are greatly appreciating your question. This is indeed an important question. In this manuscript, patients experience is process what patient undergoes, facilities to the patients and environment which patients observe at our clinic. Experience can be good or bad in any situation. While patient satisfaction is something patients feel after completing visit, which may be about the quality of services, patient will be happy and satisfied if the experience was good and unsatisfied after bad experience. We can say, feelings of satisfaction or dissatisfaction comes after good or bad experience respectively. We hope that we have made this point very clear. But if there is any confusion, we will be happy to explore it further.
- I am not surprised the response rate is only 5% in this type of survey. However, the system was revamped based on a 5% response rate. The authors need to discuss the potential bias or limitations arising from a system developed based on the feedback from a small proportion of patients.
Authors response:
Dear Professor, we are highly grateful for raising this concern and giving us chance to explore the situation. Yes, we agree with you that response rate is very low. As you know better than us that to compel someone to fill up feedback questionnaire form may not be ethical. So, we were only able to revamp, improve and optimize the system based the information’s given by the respondents in feedback. To increase the sample size of study and number of respondents, we increased the duration of study and we believe that larger sample size will minimize biasness and ambiguities in results. After revamping and optimizing system, three months data is considered as preliminary output. We have mentioned in the limitation of our study that long term data may be different from these three months output.
- The study (survey) results are not presented in the manuscript
Authors response:
Dear Professor, we are very much appreciating your positive commentary. The results of this study (survey) are presented in the result section of this manuscript. For further confirmation, I would like to indicate that figure 2, 3, 4, 5, and 7 are presenting the results of the survey.
- The authors need to show how the mobile questionnaire is designed, pretest and the process of distributed the survey. The authors did not reveal the patient profile of those who responded to the original survey, the follow up survey after revamping the system as well as the response rates of the follow up surveys. A case in point, I am not sure the data sources used to support the claims by the authors that patient satisfaction increased from 78% to 89% within just 3 months. (line310) Also, the authors should conduct the basic statistical analyses to substantiate the positive result statements.
Authors response:
Dear reviewer, we greatly appreciate your fruitful suggestions. Initially, patients’ feedback was taken on services provided to the patients before optimizations and after revamping of system, initially three months data was evaluated. Patients included this study are included are all patients visited hospital. There is possibility that if a patient visited multiple times, he might have responded to the questionnaire once or twice or even more, that is included. As far as individual profiles of patients are concerned, it can not be shared publicly. We have included the outcome and their feedback.
- Quite a few topics presented in the manuscript are opinionated contents or the description of an observed process. For example, I am not clear that the data or survey results or rationale supporting the Table 1. I am not sure how you determined the priority column in the Table.
Authors response:
Dear reviewer, we greatly appreciate your concern and your concern. In fac paper is original research and we have corrected and modified such confused places are highlighted with tracked changes. The data presented in table 1 is based on the issues identified through patients feedback and addressed appropriately.
- Describe the process of how the authors conducted the qualitative research to summarize the results. The fishbone diagram development process should be described in detail in the method section. The authors should articulate how the key waste areas and the problems were identified based on the survey results.
Authors response:
Dear reviewer, we are greatly appreciating your question, we have revised the methodology section as per your recommendations. Also, in light of editorial recommendations to keep the document at minimum word count, we have briefly mention Fishbone diagram and provided reference we followed to prepare the fishbone diagram.
Reviewer 2 Report
Patients satisfaction with fast and high-quality services is the most important in the healthcare settings and specially in outpatient departments and private clinics. High quality services on health sector are indicated by continuous quality improvement (CQI). CQI is a progressive incremental process focused on safety of all participants, outcomes, systematic process, regulated and improved working environment at the later than earlier stages. Among the various suggested strategies, we adopted the patents feedback to improve the quality of services our clinic. All of the patient visited hospital were sent a massage on mobile and requested to provide their feedback on our services. Roughly around 5% of all visitors responded and filled up questionnaire. Majority of them were satisfied with the services in various department. However, they provided us some suggestions to identify the further gaps and improve the services in improving patients experience at our clinic.
The authors considered their feedback, identified problems, redesigned the policies and implemented. After implementation of new strategies, they preliminary again evaluated the patients’ feedback on our services.
The manuscript is interesting.
It need some improvements
I have some suggestions:
1. The abstract must better proportionally summarize the sections
2. Check this part of the abstract “Patients feedback explores that optimized methods of services for the has considerably increased the patients’ satisfaction. Taken together, our this shows that patients’ feedback is very important factor to improve patients’ experience at outpatient clinics using continuous quality improvement tools. ..” Probably there is some misprinting.
3. Please insert some more details in the aims “. So, this study aims to investigate the efficiency of system at an outpatient clinic focusing on patients’ services time at various steps and obstacles faced by the patients during this encounter. Also provide some suggestions to overcome these obstacles at various steps.
4. Figure 1 must be cited and described in the body of the ms.
5. Comment the other figures.
6. Insert the labels in figure 3.
7. You have inserted a par. entitled “results and discussion”. However I think that a par. with a pure discussion is lacking . Insert this with a comparison to the literature and the limitations
Author Response
Reviewer 2
Patients satisfaction with fast and high-quality services is the most important in the healthcare settings and specially in outpatient departments and private clinics. High quality services on health sector are indicated by continuous quality improvement (CQI). CQI is a progressive incremental process focused on safety of all participants, outcomes, systematic process, regulated and improved working environment at the later than earlier stages. Among the various suggested strategies, we adopted the patents feedback to improve the quality of services our clinic. All of the patient visited hospital were sent a massage on mobile and requested to provide their feedback on our services. Roughly around 5% of all visitors responded and filled up questionnaire. Majority of them were satisfied with the services in various department. However, they provided us some suggestions to identify the further gaps and improve the services in improving patients experience at our clinic.
The authors considered their feedback, identified problems, redesigned the policies and implemented. After implementation of new strategies, they preliminary again evaluated the patients’ feedback on our services.
The manuscript is interesting. It needs some improvements. I have some suggestions:
- The abstract must better proportionally summarize the sections
Authors response:
Dear reviewer, we are very much thankful for your precious direction. In fact, to extract the further information from manuscript will increase the word count of the abstract. While respected editor has asked us to limit the abstract between 200-250 words. Considering editorial suggestions and your worthy suggestions, we have revised the abstract and have made the required changes in the abstract. We are once again indebted for your noteworthy comments.
- Check this part of the abstract “Patients feedback explores that optimized methods of services for the has considerably increased the patients’ satisfaction. Taken together, our this shows that patients’ feedback is very important factor to improve patients’ experience at outpatient clinics using continuous quality improvement tools”. Probably there is some misprinting.
Authors response:
Dear professor, we are indebted for accurate vision and reminder, we completely agree. We have found that the misprinting and revised the sentence as “Taken together, this study shows that patients’ feedback is very important factor to improve pa-tients’ experience at…”.
- Please insert some more details in the aims “. So, this study aims to investigate the efficiency of system at an outpatient clinic focusing on patients’ services time at various steps and obstacles faced by the patients during this encounter. Also provide some suggestions to overcome these obstacles at various steps.
Authors response:
Dear reviewer, we are thankful for your worthy suggestions. As mentioned in the revised manuscript, strategy used to improve the quality of services were subjected to the nature of obstacles. For instance, the issue was found at online system associated with some sort of bug, then software was optimized and bugging issue was addressed. Similarly, if there was problem human resource and education, new workers were recruited and existing workers were trained to work in more effective way and so on. These questions have been addressed in revised manuscript as needed. However, if there is still something deficient, we will revise it again under your kind suggestions. We are once again thankful for your valuable comments.
- Figure 1 must be cited and described in the body of the ms.
Authors response:
Dear reviewer, we are thankful for your reminder, figure 1 has been cited in the methodology section of manuscript at the desired place.
- Comment the other figures.
- Insert the labels in figure 3.
Authors response:
Dear professor, authors are highly grateful for your kind reminder, we have added labels in the figure 3 as per your worthy suggestions and old figure has been replaced with revised figure in the revised version of the manuscript.
- You have inserted a par. entitled “results and discussion”. However, I think that a par. with a pure discussion is lacking. Insert this with a comparison to the literature and the limitations
Authors response:
Dear reviewer, authors agree with your observation, as the paper was becoming lengthy so tried to reduce the word count in primary version of the article. As editor also emphasized to reduce the word count extensively. While considering editorial recommendations and under the light of your recommendations, we have tried to address the both questions. We have added some comparison with published literature and limitations of the study.
Round 2
Reviewer 2 Report
The ms improved.
There are not further comments.
Author Response
Tthank you very much for your time.